# Effect of Crocetin on Basal Lipolysis in 3T3-L1 Adipocytes

**DOI:** 10.3390/antiox12061254

**Published:** 2023-06-11

**Authors:** Francisco J. Cimas, Miguel Ángel De la Cruz-Morcillo, Carmen Cifuentes, Natalia Moratalla-López, Gonzalo L. Alonso, Eduardo Nava, Sílvia Llorens

**Affiliations:** 1Mecenazgo COVID-19, Regional Center for Biomedical Research (CRIB), University of Castilla-La Mancha (UCLM), 02008 Albacete, Spain; franciscojose.cimas@uclm.es; 2Food Quality Research Group, Institute for Regional Development (IDR), Campus Universitario s/n, University of Castilla-La Mancha (UCLM), 02071 Albacete, Spain; miguelangel.cruz@uclm.es; 3Regional Center for Biomedical Research (CRIB), Department of Medical Sciences, Faculty of Medicine of Albacete, University of Castilla-La Mancha (UCLM), 02008 Albacete, Spain; carmen.cifuentes@uclm.es (C.C.); eduardo.nava@uclm.es (E.N.); 4Cátedra de Química Agrícola, Higher Technical School of Agronomic and Forestry Engineering and Biotechnology (ETSIAMB), University of Castilla-La Mancha (UCLM), Campus Universitario, 02006 Albacete, Spain; nataliamoratallalopez@gmail.com (N.M.-L.); gonzalo.alonso@uclm.es (G.L.A.)

**Keywords:** crocetin, lipolysis, ATGL, HSL, perilipin-1, catalase, superoxide dismutase, adiponectin, resistin, NOS2, adipocytes, 3T3-L1

## Abstract

Crocetin (CCT) is a natural saffron-derived apocarotenoid that possesses healthy properties such as anti-adipogenic, anti-inflammatory, and antioxidant activities. Lipolysis is enhanced in obesity and correlates with a pro-inflammatory, pro-oxidant state. In this context, we aimed to investigate whether CCT affects lipolysis. To evaluate CCT’s possible lipolytic effect, 3T3-L1 adipocytes were treated with CCT10μM at day 5 post-differentiation. Glycerol content and antioxidant activity were assessed using colorimetric assays. Gene expression was measured using qRT-PCR to evaluate the effect of CCT on key lipolytic enzymes and on nitric oxide synthase (NOS) expression. Total lipid accumulation was assessed using Oil Red O staining. CCT10μM decreased glycerol release from 3T3-L1 adipocytes and downregulated adipose tissue triglyceride lipase (ATGL) and perilipin-1, but not hormone-sensitive lipase (HSL), suggesting an anti-lipolytic effect. CCT increased catalase (CAT) and superoxide dismutase (SOD) activity, thus showing an antioxidant effect. In addition, CCT exhibited an anti-inflammatory profile, i.e., diminished inducible NOS (NOS2) and resistin expression, while enhanced the expression of adiponectin. CCT10μM also decreased intracellular fat and C/EBPα expression (a transcription factor involved in adipogenesis), thus revealing an anti-adipogenic effect. These findings point to CCT as a promising biocompound for improving lipid mobilisation in obesity.

## 1. Introduction

Adipocytes store fatty acids (FAs) as triacylglycerols (TAGs) in lipid droplets (LDs) [1]. When required, these endogenous energy stores can be mobilised by hydrolysing TAGs into their constituent FAs and glycerol through the process of lipolysis. Lipolysis is carried out by the sequential actions of three enzymes: adipose tissue triglyceride lipase (ATGL), hormone-sensitive lipase (HSL), and monoglyceride lipase (MGL), with ATGL and HSL being the rate-limiting enzymes. First, ATGL hydrolyses TAG into diacylglycerol (DAG), which is converted to monoacylglycerol (MG) by HSL, and this is then cleaved by MGL into FAs and glycerol. The activation of ATGL requires the phosphorylation of its coactivator, alcohol dehydrogenase 5 (ADH5), by protein kinase A (PKA). Upon stimulation, PKA phosphorylates both HSL and perilipin-1. The phosphorylation of perilipin-1 releases ADH5 to activate and translocate ATGL to LD surfaces, where it acts to hydrolyse TAGs. Perilipin phosphorylation also causes the translocation of HSL from the cytosol to the LD surfaces and the subsequent hydrolysis of DAGs [2].

In obesity, basal adipocyte lipolysis is elevated and is directly linked to the development of insulin resistance (IR) and lipotoxicity [3]. This adipose tissue (AT) dysfunction leads to an increase in circulating FA levels. Accordingly, elevated FA levels are often observed in patients with IR and type 2 diabetes mellitus (DM2) [4]. High levels of FA, when input into the mitochondria, increases the electron transport system, resulting in the production of reactive oxygen species (ROS). The high free radical generation effect in the AT causes a dysfunctional secretory profile in this organ, resulting in a sustained overproduction of inflammatory adipocytokines that progresses to a state of obesity [5].

Free radicals, such as ROS, generally enhance the process of lipolysis in adipocytes, although they can also limit it. During inflammation in AT, ROS increase and insulin resistance is induced in the adipocyte. Thus, because insulin suppresses lipolysis, an increase in ROS levels will enhance it [6]. Antioxidants, through the scavenging of free radicals, have a critical involvement in the regulation of lipid turnover in inflammatory conditions such as obesity. In the AT, the activity of enzymatic antioxidants such as catalase (CAT), superoxide dismutase (SOD), and glutathione peroxidase (GPx) is regulated by transcriptional control [7]. In addition, many compounds from the diet are antioxidants, and these can also enhance the activity of antioxidant enzyme systems in AT [8].

Exaggerated lipolysis indicates lipotoxicity; this, together with the associated inflammatory process, due to an over-secretion of pro-inflammatory (as leptin and resistin) versus anti-inflammatory (as adiponectin) adipokines, is often seen in obese adipocytes [9,10]. These adipokines are also implicated in lipolysis modulation; thus, leptin and resistin increase lipolysis, while adiponectin decreases it [11,12].

Preventing adipocyte lipolysis may be a therapeutic approach to decrease the amount of secreted FA and could ameliorate obesity-associated lipotoxicity. Indeed, the inhibition of lipolysis in AT improves insulin sensitivity in the liver [13].

In parallel with the epidemiological rise in obesity, the intake of natural products with antioxidant properties is becoming increasingly popular and has even been used to prevent and treat metabolic disorders. These natural products can either promote or inhibit the lipolytic process [8]. In the latter case, curcumin and astragaloside (flavonoid and triterpenoid, respectively) have been shown to ameliorate IR in metabolic diseases in target organs such as the liver by decreasing lipolysis in adipose tissue [13,14]. Therefore, the use of these substances is currently being considered as an effective therapeutic strategy against lipolysis-related disorders occurring in obese conditions [8].

One of the natural products proposed as being beneficial as a therapy for obesity is crocetin. Crocetin (CCT) is an apocarotenoid present in saffron (*Crocus sativus* L.), a plant frequently used in the Mediterranean diet. CCT is highly antioxidative and shows anti-inflammatory, anti-atherosclerotic, antihypertensive, and anticancer properties [15,16,17,18]. CCT regulates the expression of pro-inflammatory and anti-inflammatory cytokines such as TNF-α and adiponectin, respectively, in adipose tissue, and it prevents visceral fat accumulation and IR [19,20]. Recently, we showed that CCT has anti-adipogenic properties [21,22]. Using the murine 3T3-L1 preadipocyte cell line, we demonstrated that CCT reduces intracellular fat in mature adipocytes by decreasing the mRNA levels of the transcription factors C/EBPβ and C/EBPα during adipogenesis [22]. Adipogenesis is the generation of new adipocytes from preadipocytes. In vitro, this process is induced by the activation of the early transcription factors CCAAT/enhancer binding protein (C/EBP), C/EBPβ, and C/EBPδ, which activate the transcription factors of late genes, namely, C/EBPα and peroxisome proliferator-activated receptor γ (PPARγ). C/EBPα and PPARγ are mutually regulated, leading to a progressive feedback mechanism that articulates the downstream transactivation of the genes related to lipid metabolism in developing and mature adipocytes. In this way, preadipocytes mature into adipocytes and promote lipogenesis and lipolysis [23].

Whether lipolysis is affected by CCT has not yet been studied. In the present study, we aimed to investigate the effect of CCT at 10 μM on basal lipolysis in 3T3-L1 adipocytes, focusing on the release of glycerol and the expression of key enzymes in the lipolysis process. We also analysed its antioxidant capacity and its influence in modulating the secretory profile of these adipocytes and investigated whether these actions are related to the antilipolytic action of CCT.

Our results show that CCT decreases glycerol release in mature adipocytes. We show that CCT diminishes the mRNA levels of ATGL and perilipin-1, increases CAT and SOD activity, enhances adiponectin expression, and reduces NOS2 and resistin expression. In addition, in mature adipocytes, CCT at 10 M in mature adipocytes decreases intracellular fat as well as the expression of C/EBPα, the transcription factor implicated in adipogenesis.

## 2. Materials and Methods

### 2.1. Plant Material and CCT

CCT was obtained from Agrícola Técnica de Manipulación y Comercialización (Minaya, Albacete, Spain) in the form of dried stigmas of saffron belonging to the protected designation of origin (PDO) “Azafrán de La Mancha”, which complies with ISO 3632:2011 (Category I). Due to its very low moisture levels, the saffron was stored in the dark at 4 °C until further use. CCT was obtained using a protected internal method from Verdú Cantó Saffron Spain (Novelda, Alicante, Spain) [18]. The preparation and reversed phase-high-performance liquid chromatography–diode array detector (RP-HPLC–DAD) analysis of CCT were performed according to [24]. CCT was provided by the Cátedra de Química Agrícola of the University of Castilla-La Mancha.

### 2.2. 3T3-L1 Cell Culture and Adipocyte Differentiation

The 3T3-L1 embryonic fibroblast cell line was acquired from ATCC (American Type Culture Collection, Manassas, VA, USA) and cultured, maintained, and differentiated according to the supplier’s instructions. Briefly, 3T3-L1 cells were expanded in 75 cm^2^ flasks at 37 °C under a humidified 5% CO_2_ atmosphere in preadipocyte Expansion Medium (EM) (Dulbecco’s Modified Eagle’s Medium (DMEM, 90%) supplemented with L-glutamine (1%), penicillin/streptomycin (0.5%), and inactivated bovine calf serum (BCS, 10%)). When the cells reached 70–80% confluence, they were seeded on 6- or 96-well or 100 mm (P100) plates and grown in Growth Medium (GM) (the same composition as EM but with BCS replaced by inactivated foetal bovine serum (FBS, 10%)) for 48 h or until the culture reached 90% confluence. Adipocyte differentiation was induced by treating post-48 h confluent cells with differentiation medium (DM) consisting of GM supplemented with an adipogenic cocktail (0.5 mM IBMX, 0.25 μM dexamethasone, and 1 μg/mL insulin (INS)). Forty-eight hours after the initiation of differentiation, the DM was replaced with adipocyte maintenance medium (MM), consisting of GM supplemented with 1 μg/mL INS, and the cells were cultured for 48 h, after which they were maintained in GM until full differentiation, 7 days after the stimulation of differentiation. Cells were considered mature adipocytes when they contained large lipid droplets. Assays were then carried out to determine the effect of CCT at 10 μM (CCT10μM) on adipogenic and lipolytic processes. Untreated cells (with CCT10μM) were the differentiation control (CTR).

Adipogenesis assays: CCT10μM was added to the 3T3-L1 cultures at the same time as the DM and the cells were incubated for 48 h. Cells were collected at the end of the differentiation process.

Lipolysis assays: After the stimulation and maintenance periods of differentiation (on day 5 post-differentiation), the 3T3-L1 adipocytes were incubated for 24 h with CCT10μM and then the medium was replaced with GM. For the positive control of lipolysis, one set of cells was stimulated with the β-adrenergic receptor agonist isoproterenol (ISO, 10 μM). At the end of the differentiation process, media were collected for the measurement of glycerol release and cells were collected to analyse the expression of lipolytic genes and to determinate antioxidant enzyme activities.

Cells were grown on (a) 6-well plates (4 × 10^4^ cells/well) to extract RNA; (b) 96-well plates were used (4 × 10^3^ cells/well) to perform the 3-(4,5-dimethylthiazol-2-yl)-2,5-diphenyltetrazolium bromide (MTT) assay and for Oil Red O staining. (c) P100 plates (2 × 10^6^ cells/P100) were used to collect cells for antioxidant activity analyses.

CCT10μM was prepared from a stock CCT solution dissolved in sterile dimethylsulfoxide (DMSO) and added to the DM. The maximal final concentration of DMSO in the cell culture was 0.001%. DMSO is usually well tolerated with no observable toxic effects to cells at a 0.1% final concentration, and it is widely used as a solvent for various pharmacological agents at concentrations of 0.05–1.5% [25]. DM+DMSO-differentiated preadipocytes were used to control the solvent.

### 2.3. MTT Assay

A 3-(4,5-dimethylthiazol-2-yl)-2,5-diphenyltetrazolium bromide (MTT) viability assay was carried out as previously described [26], with modifications. The MTT assay measures the mitochondrial activity in metabolising cells and is used as an approximate measurement of cell viability. The assay relies on the reduction of MTT, a yellow water-soluble tetrazolium dye, primarily via mitochondrial dehydrogenases, to purple-coloured formazan crystals. MTT stock in phosphate-buffered saline (PBS) was freshly prepared and assessed in all experiments. Cells were grown on 96-well sterile plates (4 × 10^3^ cells/well). At the end of the differentiation process, cells were washed with red phenol-free DMEM without FBS. Then, 100 μL of MTT solution (red phenol-free DMEM with MTT 0.5 μg/μL) was added to each well, mixed gently, and incubated for 45 min at 37 °C. Immediately, media were aspirated and discarded, and 100 μL of DMSO was added and gently stirred in each well for 3–5 min to solubilise the formazan crystals. Absorbance was determined spectrophotometrically at 570 nm using a reference wavelength of 630 nm (ASYS UVM 340, Cambridge, United Kingdom, Microplate Readers). The colour intensity is directly proportional to the number of viable cells. Data obtained from at least 10 replicates of each experimental condition across three independent experiments were used for analysis. Appropriate solvent controls were included.

### 2.4. Glycerol Assay

The cultured medium was collected to determine the extracellular glycerol content using a Free Glycerol Assay kit (ab65337, Abcam, Cambridge, United Kingdom) according to the protocol for the colorimetric assay provided by the manufacturer. Briefly, 2 × 10^6^ snap frozen cells (−80 °C) were resuspended and homogenised in 500 μL of ice-cold glycerol buffer. After centrifugation (15 min at 4 °C at 10,000× *g*), the supernatant was collected. Several dilutions of the sample were performed using the Glycerol Assay Buffer. A standard curve was prepared from 1 mM Glycerol Standard provided in the kit. Samples and the standard curve were mixed with Reaction Mix (containing a glycerol probe, glycerol enzyme mix, and Glycerol Assay Buffer) in a clean 96-well plate. Plates were incubated at 37 °C for 30 min and protected from light. In the assay, glycerol is enzymatically oxidised to generate a product that reacts with the probe to generate colour. Finally, absorbance was determined spectrophotometrically at 570 nm (ASYS UVM 340, Cambridge, United Kingdom, Microplate Readers).

### 2.5. Total RNA Isolation and Quantitative Real-Time PCR Analysis

For each experimental condition, RNA was extracted from 3T3-L1 cells and prepared using the PureLink™ RNA Mini Kit (Thermo Fisher Scientific Inc., Waltham, Massachusetts, USA) and the RNeasy Mini Kit (50974104, Qiagen, Hilden, Germany) according to manufacturers’ instructions. RNA from undifferentiated preadipocytes was also extracted to compare the expression levels of different genes. The quantity and integrity of the extracted RNA were verified and quantified spectrophotometrically using NanoDrop (Thermo Fisher Scientific Inc., Waltham, Massachusetts, USA). Complementary DNA (cDNA) was synthesised from 1 μg of RNA using a RevertAid H Minus First Strand cDNA synthesis kit (Thermo Fisher Scientific Inc., Waltham, Massachusetts, USA) according to the manufacturer’s protocol. Gene expression was assessed using quantitative real-time polymerase chain reaction (qRT-PCR) in a LightCycler 480 II or 7500 Fast thermocyclers using Fast Sybr Green Master Mix (Applied Biosystems, Thermo Fisher Scientific Inc., Foster City, California, USA). We measured β-2-microglobulin (in the case of lipolysis analyses) and β-actin (in the case of the adipogenesis analyses) as endogenous controls. The primer sequences used for amplification are presented in Table A1, Appendix A.

The reaction mixtures were incubated for an initial denaturation at 95 °C for 10 min, followed by 45 PCR cycles of 95 °C for 15 s, 60 °C for 1 min, 95 °C for 15 s, and 60 °C for 1 mine. The ΔΔCT method was used for relative quantification, and the levels of transcripts were normalised to β-actin or β-2-microglobulin. The mRNA expression levels were calculated relative to the basal condition in which the 3T3-L1 cells were treated for 48 h with the adipogenic cocktail but without CCT treatment (CTR). Three independent experiments were performed, each in triplicate. Fold changes of gene expression were calculated using the 2^−ΔΔCt^ method [27].

### 2.6. Determination of Antioxidant Enzyme Activity

#### 2.6.1. Catalase (CAT) activity

Catalase (CAT) activity assays were performed using a colorimetric CAT activity assay kit (ab83464, Abcam, Cambridge, United Kingdom), whereby the CAT present in a sample reacts with H_2_O_2_ to produce water and oxygen. The unconverted H_2_O_2_ reacts with a probe to produce a product that can be measured colorimetrically. Cells from the P100 plates (10^6^/P100) were collected, washed with cold PBS, and then resuspended and homogenised in 200 μL of ice-cold assay buffer. After centrifugation (15 min at 4 °C at 10,000× *g*), the supernatant was collected and mixed with fresh 1 mM hydrogen peroxide (H_2_O_2_) solution and assay buffer in a clean 96-well plate. After incubation at room temperature for 30 min, stop solution was added to each sample. Then, Developer Mix (containing OxiRed probe and horseradish peroxidase (HRP) solution) was added to each reaction. Plates were incubated at room temperature for 10 min and protected from light. Finally, the absorbance was determined spectrophotometrically at 570 nm (ASYS UVM 340, Cambridge, United Kingdom, Microplate Readers). All experiments were performed in duplicate with three independent experiments.

#### 2.6.2. Superoxide Dismutase (SOD) Activity

In the SOD assay, SOD catalyses the dismutation of the superoxide anion produced by the action of the xanthine oxidase (XO) into H_2_O_2_ and superoxide anions. The superoxide anions react to produce a water-soluble formazan dye that can be detected using colorimetry. The greater the activity of SOD in the sample, the lower the amount of formazan dye produced. Cells from the P100 plates (2 × 10^6^/P100) were collected and lysed in ice-cold 0.1 M Tris/HCl, pH 7.4, containing 0.5% Triton X-100, 5 mM β-mercaptoethanol, and 0.1 mg/ml phenylmethanesulfonyl fluoride (PMSF). After centrifugation (15 min at 4 °C at 14,000× *g*), the supernatant was collected and transferred to 96-well plates, and SOD activity was measured using the Superoxide Dismutase Activity Assay Kit (ab65354, Abcam, Cambridge, United Kingdom). Absorbance was determined spectrophotometrically at 450 nm (ASYS UVM 340, Cambridge, United Kingdom, Microplate Readers) following incubation at 37 °C for 20 min. All experiments were performed in duplicate with three independent experiments. In this kit, one unit is defined as the amount of SOD that inhibits XO activity by 50% under the assay conditions.

#### 2.6.3. Glutathione Peroxidase (GPx) Activity

In the GPx assay, GPx oxidises glutathione (GSH) to produce glutathione disulphide (GSSG) as part of the reaction in which it reduces cumene hydroperoxide. Glutathione reductase then reduces the GSSG to produce GSH while concomitantly consuming nicotinamide adenine dinucleotide phosphate (NADPH). The decrease in NADPH (measured at 340 nm) is proportional to GPx activity. Cells from the P100 plates (2 × 10^6^/P100) were collected, and GPx activity assays were performed using the colorimetric GPx activity assay kit (ab102530, Abcam, Cambridge, United Kingdom). The collected cells were washed with ice-cold PBS and lysed with 200 µL of cold assay buffer. Following centrifugation (15 min at 4 °C at 10,000× *g*), the supernatants were transferred to clean tubes. Then, 15–20 µL samples were added to the reaction mixture (40 mM NADPH solution, glutathione reductase, and glutathione) in 96-well plates. After incubating at room temperature for 15 min, 10 µL of cumene hydroperoxide solution was added to the reaction system. The concentration of NADPH was measured using a microplate reader at 340 nm. Measurements were performed at 0 and 7 min after the addition of cumene hydroperoxide and samples were protected from light and incubated at 25 °C for 5 min between measurements. GPx activities were calculated from the changes in NADPH concentrations. All experiments were performed in duplicate with three independent experiments.

### 2.7. Oil Red O Measurement, OR

Cells differentiated in 96-well sterile plates were stained with Oil Red O (OR) at the end of the differentiation process according to standard procedures [28]. Oil Red O is a dye that strongly stains lipids, specifically triacylglycerols, and it is often used for the quantitative analysis of adipocyte differentiation [28]. The OR stock solution was prepared one day prior to its use as follows: 0.2 g of OR was dissolved in 100 mL of isopropanol for 24 h at room temperature under agitation. The OR working solution was prepared by mixing six parts of OR stock solution and four parts of ddH_2_O and filtered through two layers of Whatman filter paper to remove any precipitate. For OR staining, the cells were first washed with PBS three times and fixed in 4% formaldehyde for 1 h at room temperature, avoiding any shaking of the plate. The formaldehyde was removed, and the cells were washed once with cold PBS and air-dried for 10 min. Freshly prepared OR working solution was added to cover the cell surface. After 10 min, the solution was aspirated, and cells were washed 3 times with cold PBS and air-dried for 15 min. OR was eluted with 100% isopropanol for 10 min, and the absorbance at 450 nm was measured using a spectrophotometer (ASYS UVM 340, Cambridge, United Kingdom, Microplate Readers). Data obtained from at least 10 replicates of each condition across three independent experiments were used for analysis. The amount of colour produced is directly proportional to the amount of intracellular fat. Absorbance measures were also taken from wells devoid of cells or only containing the OR in order to subtract any colour that may have been adsorbed by the plastic walls of the well.

All reagents were acquired from Sigma–Aldrich unless specified otherwise.

### 2.8. Data Analysis

All data are presented as mean ± standard deviation (SD) of three independent experiments. One-way analysis of variance (ANOVA) and a post hoc Bonferroni’s multiple-comparison test, using GraphPad Prism version 5.0 software, were used to identify differences between groups. The statistical significance of mRNA levels was evaluated using Student’s *t*-test in GraphPad Prism v7.0 software. Results with a *p*-value < 0.05 were considered significant.

## 3. Results

### 3.1. CCT Diminishes Glycerol Release without Altering the Viability of Cells in Culture

First, we tested whether CCT at 10 μM produced cytotoxicity in the cells. To rule out the cytotoxicity of CCT10μM, cell viability assays were conducted in the presence of the compound. An MTT assay was used as an indicator of cytotoxicity at the end of the differentiation process. Neither CCT nor its solvent, DMSO, significantly reduced cell viability relative to the untreated control cells, CTR, (105.1% and 100%, respectively) (Figure 1, panel A). Next, we investigated the effects of CCT10μM on the basal lipolytic process in the 3T3-L1 murine preadipocyte cell line. The most direct assessment of lipolysis is the measurement of the lipolytic products released by adipocytes (FAs and glycerol) into the incubation media, which, under basal conditions, occurs in a molar ratio close to 1:1 (FAs/glycerol) [29]. Glycerol release was determined at the end of the differentiation process. Isoproterenol (ISO) was used as a positive control of lipolysis. We found that a 24 h incubation of 3T3-L1 adipocytes (5 days post-differentiation) with CCT10μM significantly reduced the glycerol release (75%) compared with that from CTR cells and cells treated with ISO (*p* < 0.05) (Figure 1, panel B). As expected, incubation with ISO (10 μM) produced a significant increment in glycerol release compared with the CTR cells (128%, *p* < 0.05). The effect of DMSO, used as solvent of CCT, was also evaluated to account for possible solvent effects. The results showed no significant effect compared with the CTR cells (97.5%, see below). These findings indicate that CCT10μM produces a decrease in glycerol production.

These results suggest that CCT may modulate basal lipolysis without altering the viability of cells in culture.

### 3.2. CCT Diminishes Adipose Triglyceride Lipase (ATGL) and Perilipin-1 Expression

Next, we used qRT-PCR to analyse the effect of CCT10μM on the mRNA expression of two key lipolytic enzymes, adipose triglyceride lipase (ATGL) and hormone-sensitive lipase (HSL). Additionally, we evaluated the effect of CCT on the mRNA expression of two proteins implicated in lipolysis, ADH5 and perilipin-1. We found that a 24 h incubation of 3T3-L1 adipocytes (5 days post-differentiation) with CCT10μM significantly reduced ATGL mRNA expression by 22% (*p* < 0.001); however, no significant reduction effect was found for its coactivator, ADH5 (Figure 2, panels A and C). On the other hand, this treatment did not significantly affect the mRNA expression of HSL, but it significantly reduced that of perilipin-1 by 26.3% (*p* < 0.05). In contrast, although ISO treatment tended to decrease ATGL (10.5%), HSL (20%), and perilipin-1 (9.8%) and increase ADH5 (23%) mRNA levels, these changes were not statistically significant. These results suggest that CCT can modulate lipolysis in 3T3-L1 adipocytes by selectively inhibiting key lipolytic enzymes.

### 3.3. CCT Increases the Activity of Catalase (CAT) and Superoxide Dismutase (SOD)

Dysregulation of lipolysis is affected by an increase in ROS. Thus, antioxidant enzymes play an important role in modulating lipolysis by neutralising these species. Many compounds in the diet exhibit the ability to enhance the activity of antioxidant enzyme systems. CCT is a powerful antioxidant biocompound; hence, we evaluated the effect of CCT10μM on the activity of the major antioxidant enzymes catalase (CAT), superoxide dismutase (SOD), and glutathione peroxidase (GPx) in differentiated 3T3-L1 adipocytes. We treated cells with CCT10μM for 24 h on day 5 post-differentiation and measured the activity of CAT, SOD, and GPx. We found that CCT10μM significantly increased the activity of CAT (118%) compared with the use of control conditions (CTR) or treatment with ISO (*p* < 0.05). ISO treatment appeared to slightly reduce CAT activity; however, this was not significant (89%) (Figure 3, CAT). The effect of DMSO on CAT activity was also found not to be significant relative to control cells (107%). With regard to SOD activity, this was significantly increased after incubation with CCT10μM compared with the control (215%) and with respect to DMSO (*p* < 0.05) and ISO (*p* < 0.05) (Figure 3, SOD). In this case, both DMSO and ISO showed a non-significant increasing trend (161% and 146%, respectively, compared to the control). CCT10μM, DMSO, and ISO did not elicit any detectable effect on GPx activity (102%, 95%, 97%, respectively, compared with control conditions) (Figure 3, GPx). These results suggest that CCT can enhance CAT and SOD activities, which may contribute to its antioxidant effects by scavenging hydrogen peroxide and superoxide anions in basal lipolysis.

### 3.4. CCT Diminishes Inducible Nitric Oxide Synthase (NOS2) Expression

Some authors have reported that lipolysis can be modulated using nitric oxide (NO) and that NO donor drugs increase basal lipolysis in adipocytes [30]. It was therefore of interest to test whether the 24 h incubation (day 5 post-differentiation) of 3T3-L1 adipocytes with CCT10μM could modulate NOS2 and NOS3 mRNA levels. We found that, under these conditions, CCT10μM did not modify NOS3 levels but strongly reduced NOS2 mRNA expression (36.9%, *p* < 0.01) (Figure 4). Incubation with ISO produced a significant increase (82.5%, *p* < 0.001) in NOS2 expression. However, although ISO incubation showed a tendency to decrease NOS3 (25%) mRNA levels, this change was not statistically significant (Figure 4).

### 3.5. CCT Enhances Adiponectin but Diminishes Resistin Expression

Adipokines are cytokines secreted by adipose tissue that are able to regulate lipid and glucose metabolism [31] and modulate the rate of lipolysis [11,32]. We measured the mRNA levels of adiponectin (ADIPOQ), resistin (RETN), and leptin (LEP) using qRT-PCR in differentiated 3T3-L1 adipocytes treated for 24 h with CCT10μM on the fifth day post-differentiation. We found that CCT10μM significantly increased adiponectin mRNA expression four-fold compared with the control (*p* < 0.001) and reduced resistin mRNA levels by 50% (*p* < 0.001). No effect on leptin expression levels in these cells was detected (Figure 5). Treatment with ISO decreased both adiponectin and resistin mRNA levels (71% *p* < 0.01 and 79% *p* < 0.01, respectively) but did not modify those of leptin. These results suggest that CCT modulates the expression of adipokines, enhancing levels of the anti-inflammatory and insulin-sensitising adiponectin and decreasing levels of the pro-inflammatory and insulin-resistance-promoting resistin.

### 3.6. CCT Reduces Intracellular Fat and Diminishes C/EBPα Expression

In previous studies, we reported that CCT5μM, but not CCT1μM, was able to reduce intracellular fat by modifying the expression of key genes related to adipogenesis [22]. In this study, we wanted to investigate whether the stimulation of differentiation in 3T3-L1 adipocytes with CCT at 10 μM affects both the content of intracellular fat and the expression of late genes involved in adipogenesis. First, we stimulated the differentiation of preadipocytes in the presence of CCT10μM for 48 h. The amount of intracellular fat was determined using OR staining at the end of the differentiation process. We found that the incubation of CCT10μM together with an adipogenic cocktail for 48 h significantly decreased the content of intracellular fat by 12% (82%, *p* < 0.05) compared with the differentiation control (CTR). In addition, this reduction was significantly different relative to the effect of its solvent DMSO (*p* < 0.05). No significant effect on the content of intracellular fat was produced by DMSO (101%) compared with CTR (Figure 6, panel A). Using qRT-PCR, we also quantified and analysed the effect of CCT10μM on the expression of a pair of late genes of adipogenesis, PPARγ and C/EBPα, at the end of the differentiation process. We found that the incubation of CCT10μM together with the adipogenic cocktail for 48 h produced a significant reduction in C/EBPα mRNA levels compared with the untreated control cells (CTR) (16.4%, *p* < 0.05); however, this did not affect PPARγ mRNA levels (Figure 6, panel B).

Thus, this result indicates that CCT10μM produces a decrease in intracellular fat and inhibits adipocyte differentiation by downregulating the expression of C/EBPα, as has been demonstrated in previous studies.

## 4. Discussion

Lipolysis is a crucial process that provides organisms with FAs and glycerol during periods of negative energy balance via the hydrolysis of TAGs. This process is fundamentally triggered by β-adrenergic stimulation; however, a basal lipolytic state exists [33,34]. Obesity is most frequently associated with disturbances in lipolysis, especially in basal lipolysis. In this metabolic condition, the basal rate of lipolysis is elevated, and this strongly contributes to the development of insulin resistance and lipotoxicity [3,35]. While FAs play a fundamental role in supplying energy demand, over-supplementation is highly detrimental.

Carotenoids are implicated in adipose tissue biology [36], and numerous studies have shown that the natural apocarotenoid CCT could play a preventive or therapeutic role in some aspects related to the comorbidities that accompany obesity. Indeed, CCT has been shown to prevent visceral fat accumulation and insulin resistance induced by a hypercaloric diet in rats [20]. Thus, in the present study, we tested the anti-lipolytic effects of CCT, using glycerol concentration as an index for lipolysis [29]. The glycerol content in the incubation media was measured after treatment with CCT10μM for 24 h. Our results showed that CCT10μM decreased basal lipolysis without affecting cellular viability. Thus, we investigated the possible implication of CCT in lipolysis mechanisms by studying its effect on the expression of key lipolytic enzymes.

In adipose tissue, ATGL and HSL are responsible for more than 90% of TAG hydrolysis [34]. As mentioned in the introduction section, the activation of ATGL by its coactivator ADH5 and phosphorylation of HSL and perilipin-1 by protein kinase A (PKA) are the main mechanisms of lipolysis regulation.

In the present work, we found that treatment with CCT10μM diminished the expression of ATGL and perilipin-1 while that of HSL and ADH5 remained unchanged. Our results suggest that CCT inhibits basal lipolysis by downregulating ATGL, and this may explain the reduction in the glycerol content of differentiated 3T3-L1 adipocytes. It is known that ATGL plays a critical role as a regulator of basal lipolysis. Animal studies have shown that the absence or silencing of ATGL decreases basal lipolysis [34,37]. Furthermore, ATGL overexpression increases the basal lipolytic capacity of adipocytes in humans and mice. In contrast, HSL overexpression or silencing does not affect basal lipolytic rates in human adipocytes [38,39]. While these findings are in line with our results, further experiments are necessary to clarify the mechanisms by which CCT downregulates ATGL. It has been reported that several transcription factors which drive adipogenesis also regulate ATGL transcription, and an example of this is C/EBPα [40]. CCT at 10 μM downregulates C/EBPα (see the last part of Section 4). Therefore, it is plausible that CCT diminishes ATGL expression via C/EBPα downregulation.

For HSL to be fully activated it must gain access to LDs. In adipocytes, this is mediated by perilipin-1. Perilipin-1 is a lipid droplet scaffold protein that plays a central role in coordinating interactions among lipolytic enzymes. The phosphorylation of perilipin-1 results in conformational changes that expose TAGs stored in LDs and facilitates the translocation of HSL to the LD, thus increasing the lipolytic process [41]. In the present study, we found that CCT10μM downregulates perilipin-1. The reduction in perilipin-1 expression together with the diminished glycerol release elicited through CCT10μM treatment surprised us since it has been demonstrated that perilipin-null mice exhibit elevated basal lipolysis [42]. A possible explanation for our results is that, while the absence or reduction of perilipin-1 would leave TAGs within LDs exposed to hydrolysis by HSL, this hydrolase is not able to hydrolyse TAGs properly. HSL possesses a ten-fold higher affinity for DAGs compared to TAGs, making it less efficient than ATGL in breaking down TAGs [34]. In fact, HSL-deficient mice display DAG accumulation in several tissues [43], suggesting that, although HSL may catalyse TAG hydrolysis, the major physiological substrates are DAGs and not TAGs [37]. In addition, it is known that, without perilipin-1, HSL is unable to access the LD and that the binding of HSL to the droplets’ surface occurs at a very low level [42]. Although CCT decreases perilipin-1 levels, the glycerol concentration found in our work is lower than expected, possibly due to the decrease in ATGL levels.

It is accepted that an increased FA flux from hypertrophic adipose tissue to insulin-sensitive tissues, which occurs in obesity or DM2, contributes to insulin resistance and glucose intolerance [44]. Accordingly, numerous studies have confirmed that a reduced FA mobilisation from adipose tissue (either with global or adipocyte-specific deletion of ATGL or HSL) leads to a significantly increased and improved insulin sensitivity [43,45]. Interestingly, HSL deficiency in humans leads to insulin resistance [40]. Therefore, by downregulating ATGL, CCT may improve insulin sensitivity.

An intracellular redox environment is fundamental for several biological functions, balancing the production of pro-oxidant factors, such as reactive oxygen species (ROS), and the antioxidant enzymatic system (CAT, SOD, and GPx) [7]. Thus, oxidative stress develops upon excessive ROS production or the impairment of the antioxidant response [46]. ROS have been shown to be important induction factors of lipolysis [47]. During lipolysis, an increase in ROS levels generated by mitochondrial activity takes place, although this can be buffered by the activation of antioxidant defences. This occurs as follows: SOD dismutates superoxide anions into the strong oxidant H_2_O_2_, which is subsequently transformed into water by CAT and GPx, peroxiredoxin, or thioredoxins. In obesity, both ROS and lipolysis are increased, and this elevated production of ROS depletes the antioxidant systems and causes oxidative stress [48,49]. A recent review on the role of free radicals and antioxidants regulating lipolysis and lipogenesis in AT reported that the antioxidant enzyme activities of GPx and SOD are dysregulated in obese individuals, and that SOD, GPx, and CAT are significantly downregulated in the abdominal adipose tissue of metabolic syndrome female rats [50]. A number of studies support the idea that certain natural antioxidant biocompounds may serve as ROS scavengers and/or enhance the activity of antioxidant enzymes [51,52]. CCT has been shown to have both antioxidant and anti-inflammatory properties [15,21,53,54]. It has been shown that CCT is capable of lowering the levels of oxidative stress biomarkers such as malondialdehyde (MDA) [55,56]. In fact, a recent publication has shown that CCT (from saffron) at the same concentration as the one we used (10 μM) is the most effective dose for reducing MDA levels [56]. In the present study, we investigated the ability of CCT to enhance antioxidant enzyme activities during basal lipolysis, specifically those of CAT, SOD, and GPx. The antioxidant capacity or mRNA expression levels of these antioxidant enzymes have been reported to be decreased in obese humans or mice [48]. In the present work, we found that CCT increases CAT and SOD activities while not affecting the activity of GPx. This antioxidant activity of CCT could partly explain CCT’s inhibition of basal lipolysis. In line with these findings, human adipocytes treated with another natural biocompound, resveratrol, also reduces ROS concentration and results in an inhibition of lipolysis [47].

According to the literature, the antioxidant mechanism by which lipolysis is reduced involves the targeting of HSL to prevent its phosphorylation [57,58,59]. It is unknown whether ROS are able to modify the structure of ATGL or whether ROS act as co-activators of this enzyme. Some studies have suggested that ATGL may have intrinsic antioxidant activity since ATGL-deficient mice tend to show increased oxidative stress [60].

On the other hand, GPx was not affected by the CCT10μM treatment during basal lipolysis. It is known that GPx is an antioxidant enzyme that reduces H_2_O_2_ to water, protecting cells against lipid peroxidation. Similar to that of the other enzymes, its expression in adipocytes is dysregulated in human obesity and metabolic disorders [61,62]; however, a direct role for GPx in the lipolysis process has not yet been established.

Adipocytes express two members of the NOS family, NOS2 and NOS3 [63]. NOS2 is responsible for the high NO production in infections or inflammatory diseases. These isozymes are implicated in the comorbidities associated with obesity through NO production, which has essential functions including regulating adiposity, energy expenditure, and insulin sensitivity [64]. It has been reported that the NOS family of enzymes is dysregulated in isolated fat cells and adipose tissue sections derived from obese individuals, at both the mRNA and protein levels [65,66]. NOS3 is associated with anti-obesogenic effects, while NOS2 promotes insulin resistance [64]. We found that CCT did not modify NOS3 but diminished NOS2 expression. Although our experiments were not performed under inflammatory conditions, CCT showed anti-inflammatory properties by selectively downregulating NOS2 and, therefore, diminishing the NO produced by this source. Thus, our results reinforce the notion that the NO system contributes to the antilipolytic and anti-inflammatory actions of CCT.

NOS2 generates NO at a high rate and for prolonged periods, which favours a shift in the cellular redox potential to a more oxidised state. Evidence reveals that low concentrations of NO, such as would result from CCT downregulating NOS2, protect against ROS-associated injury [67]. Reports show that NO restrains lipolysis via the oxidative modification of adenylyl cyclase (AC), mimicking antioxidant actions, and it has been suggested that NO suppresses lipolysis by reducing cyclic adenosine monophosphate (cAMP) and PKA activation [30,68], avoiding HSL and perilipin-1 phosphorylation. Thus, these authors established that the regulatory role of NO in lipolysis is linked to the redox state of NO. It is known that NO activates soluble guanylyl cyclase (sGC) [69]; however, the lipolytic effect of sGC is unknown.

Our results show that CCT10μM increases the expression of adiponectin while diminish that of resistin but does not modify leptin mRNA expression in basal lipolysis. Adipokines including adiponectin, resistin, and leptin have important roles in regulating lipid and glucose metabolism and adipose tissue biology [31,70]. Low levels of adiponectin are expressed in enhanced adiposity, while those of resistin and leptin have been found to be increased in obesity [71,72]. It has also been postulated that resistin is linked to obesity, insulin resistance, and diabetes [73]. Consequently, the suppression of leptin and resistin expression may be an alternative means of combating obesity. Phytochemicals have been reported to affect the mRNA expression of multiple adipokines. Moreover, resveratrol reduces leptin levels and increases adiponectin expression [74,75]. Anthocyanins increase adiponectin expression in human adipocytes [76]. Additionally, CCT can regulate the expression of adiponectin in the adipose tissue of fructose-fed rats [77]. In line with these natural compounds, CCT increases adiponectin expression in 3T3 adipocytes.

While we did not analyse the implications of adiponectin levels during basal lipolysis in this study, the antioxidant activity of CCT could be related to the elevated levels of adiponectin. In line with this, reports have shown that high ROS levels lead to a decrease in adiponectin expression and secretion [78].

Resistin has been associated with a higher rate of lipolysis by enhancing the transcription of ATGL and HSL [79]. In this study, we showed that CCT produced a reduction in resistin expression during basal lipolysis. Thus, the effects of resistin could be related to the ability of CCT to diminish basal lipolysis.

Our results show that CCT did not modify LEP expression. As mentioned in the introduction section, it has been reported that leptin increases lipolysis in vitro [12]. However, the predominant effects of leptin in enhancing lipolysis in adipose tissue are mainly mediated by the peripheral nervous system increasing the sympathetic efferent signal [80].

An altered adipogenic process and increased lipid accumulation are key features in obesity. In this context, the control of adipogenesis has also emerged as a potential target for the prevention and/or treatment of obesity [81]. We have previously reported that CCT5μM, but not CCT1μM, reduces lipid accumulation [22]. Since different concentrations of CCT trigger diverse results, in this study, we aimed to investigate the effect of CCT at 10 μM on the adipogenic process. Our results show that CCT10μM, added at the early stage of cell differentiation, diminished the intracellular content of TAGs, showing an anti-adipogenic effect. We also analysed the effect of CCT10μM on the expression of C/EBPα and PPARγ, which are late genes that are important in adipogenesis. CCT10μM downregulated the C/EBPα transcription factor at the end of the differentiation process, while PPARγ expression was not modified. These results confirm our previous result obtained with CCT at 5 μM [22].

In summary, CCT is a candidate antilipolytic and anti-adipogenic molecule. Our study provides the first direct evidence that the antilipolytic action of CCT in adipocytes may allow this biocompound to lower levels of circulating FAs, which would be helpful in disorders such as obesity and DM2. Future research should aim to investigate the phosphorylation state of HSL and perilipin-1 and to clarify the mechanisms by which CCT downregulates ATGL. In addition, further studies are required to evaluate the effect of CCT on lipolysis under inflammatory conditions.

## 5. Conclusions

CCT10μM reduces the content of glycerol, showing an antilipolytic capacity by downregulating both ATGL and perilipin-1 without modifying HSL expression levels. The downregulation of C/EBPα and resistin by CCT10μM could be involved in decreasing ATGL expression.

The antioxidant properties of CCT10μM might underlie this antilipolytic activity by upregulating both CAT and SOD and downregulating NOS2 expression. This would prevent an increase in ROS and therefore prevent ROS-induced lipolysis. In addition, CCT10μM upregulates adiponectin, which is known to reduce lipolysis. The increase in these antioxidant mechanisms could explain both the increase in adiponectin by CCT10μM and even the fact that HSL cannot fully carry out its enzymatic activity (probably by preventing its phosphorylation). The latter might be also related to both the NO released by NOS3, which was not affected by CCT10μM, and the decrease in perilipin-1 elicited by CCT10μM.

Furthermore, the upregulation of adiponectin and downregulation of both resistin and NOS2 expression by CCT10μM strongly suggest that this compound possesses anti-inflammatory actions. Altogether, these actions, which are promoted by CCT, contribute to the diminishing of the deleterious effects of elevated lipolysis that lead to a pro-oxidant and pro-inflammatory state in obesity.

## Figures and Tables

**Figure 1 antioxidants-12-01254-f001:**
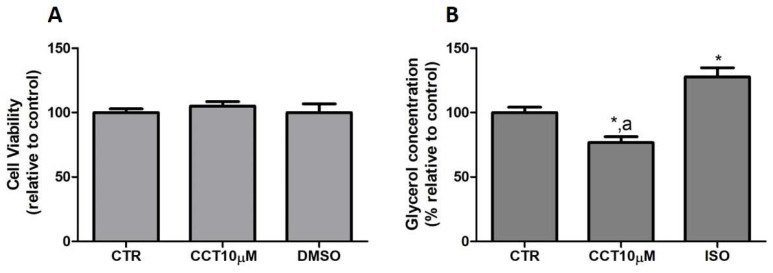
(**A**) Cell viability. Five-day post-differentiation preadipocytes were incubated with CCT10μM for 24 h. Cells were collected at the end of the differentiation process and their associated mitochondrial metabolic activity values were measured using the MTT assay. Based on these values, the cell viability of differentiated cell cultures treated with CCT10μM is expressed as a relative percentage of control differentiation cultures (CTR). (**B**) Glycerol content. Five-day post-differentiation preadipocytes were incubated with CCT10μM for 24 h. Glycerol values were determined at the end of the differentiation process. Glycerol release from differentiated cell cultures treated with CCT10μM or ISO (10 μM) is expressed as a percentage relative to control differentiation conditions (CTR). Data were obtained from at least 3 replicates of each condition and from three independent experiments. Data are expressed as mean ± SD. * and ^a^ indicate a significant difference compared with control differentiation or ISO conditions, respectively (*p* < 0.05).

**Figure 2 antioxidants-12-01254-f002:**
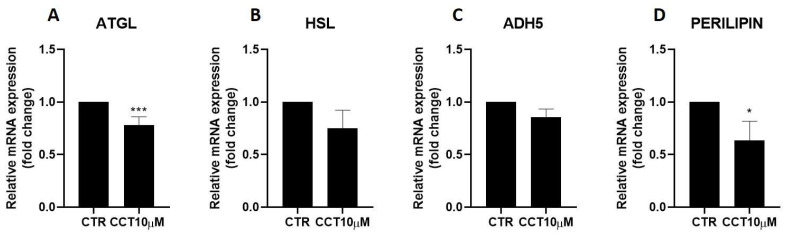
mRNA expression levels of key lipolytic enzymes. Five-day post-differentiation preadipocytes were incubated with CCT10μM for 24 h. The mRNA expression levels of (**A**) adipose triglyceride lipase (ATGL), (**B**) hormone-sensitive lipase (HSL), (**C**) alcohol dehydrogenase 5 (ADH5), and (**D**) perilipin-1 at the end of the differentiation process were evaluated via qRT-PCR using specific primer pairs. Relative qRT-PCR values were corrected against β-2-microglobulin expression levels and normalised with respect to the differentiation control (CTR). Data were obtained from at least 3 replicates of each condition from three independent experiments. Data are expressed as mean ± SD. For each gene, the maximum mRNA expression level at the end of the differentiation process for the untreated control cells (CTR) was set at 1, and the relative mRNA expression level in cells treated with CCT10μM at the same time point is depicted. * and *** indicate a significant difference in *p* < 0.05 and *p* < 0.001, respectively, compared with control differentiation.

**Figure 3 antioxidants-12-01254-f003:**
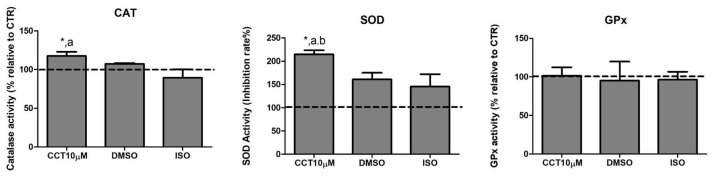
Antioxidant system activity. Five-day post-differentiation preadipocytes were incubated with CCT10μM for 24 h. The antioxidant activity of catalase (CAT), superoxide dismutase (SOD), and glutathione peroxidase (GPx) was measured at the end of the differentiation process. Values are expressed as percentages relative to cells under control differentiation conditions (CTR, dashed line). Data were obtained from at least 2 replicates of each condition from three independent experiments. Data are expressed as mean ± SD. *, ^a^, and ^b^ indicate a significant difference in *p* < 0.05 compared with control differentiation, ISO, and DMSO treatments, respectively.

**Figure 4 antioxidants-12-01254-f004:**
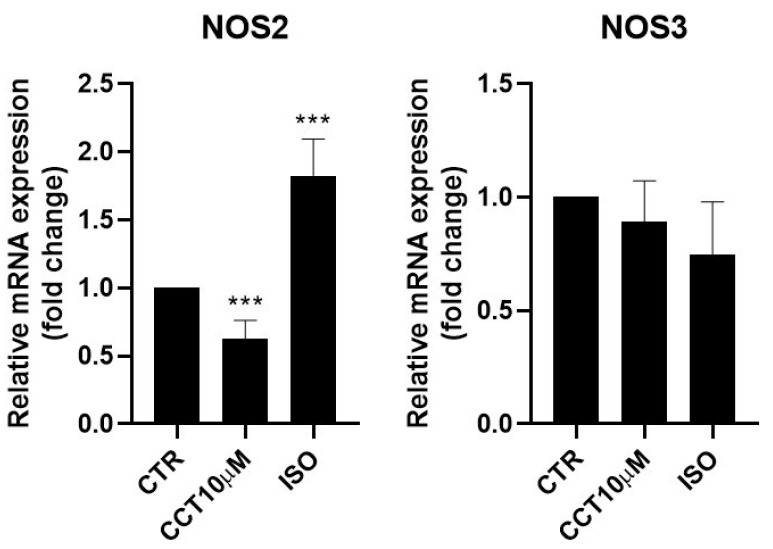
mRNA expression levels of nitric oxide synthases. Five-day post-differentiation preadipocytes were incubated with CCT10μM for 24 h. The expression of inducible and endothelial nitric oxide synthase (NOS2 and NOS3, respectively) was evaluated using qRT-PCR with specific primer pairs at the end of the differentiation process. The relative qRT-PCR values were corrected against β-2-microglobulin expression levels and normalised with respect to control differentiation conditions (CTR). Data were obtained from at least 3 replicates from three independent experiments. Data are expressed as mean ± SD. The maximum mRNA expression levels at the end of the differentiation process for the untreated control cells (CTR) were set at 1, and relative mRNA expression levels for cells treated with CCT10μM at the same time point are depicted. *** indicates a significant difference in *p* < 0.001, compared with control differentiation.

**Figure 5 antioxidants-12-01254-f005:**
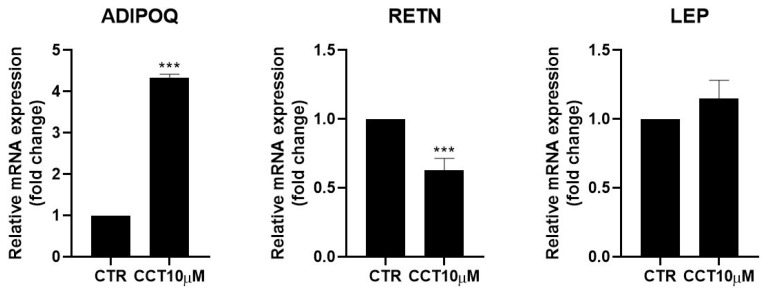
mRNA expression levels of adipokines. Five-day post-differentiation preadipocytes were incubated with CCT10μM for 24 h. The expression of adiponectin, resistin, and leptin (ADIPOQ, RETN, and LEP, respectively) was evaluated using qRT-PCR with specific primer pairs at the end of the differentiation process. The relative qRT-PCR values were corrected against β-2-microglobulin expression levels and normalised with respect to control differentiation conditions (CTR). Data were obtained from at least 3 replicates from three independent experiments. Data are expressed as mean ± SD. The maximum mRNA expression levels at the end of the differentiation process for the untreated control cells (CTR) were set at 1, and relative mRNA expression levels for cells treated with CCT10μM at the same time point are depicted. *** indicates a significant difference compared with control differentiation (*p* < 0.001).

**Figure 6 antioxidants-12-01254-f006:**
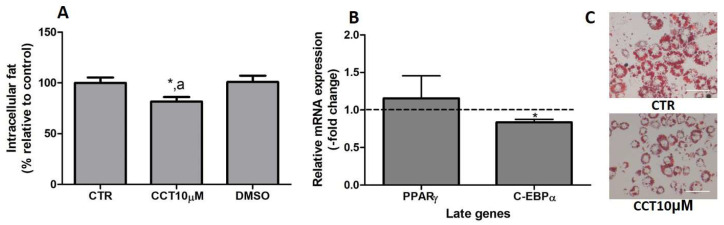
(**A**) Intracellular fat. 3T3-L1 adipocytes were treated with an adipogenic cocktail together with CCT10μM for 48 h to induce differentiation. Intracellular fat values were determined at the end of the differentiation process and are expressed as percentages relative to control differentiation conditions (CTR). The effect of the CCT solvent DMSO (<0.001% final concentration in culture media) was also compared. (**B**) mRNA expression of two late genes of adipogenesis, PPARγ and C/EBPα. Two-day post-confluence preadipocytes were incubated with CCT10μM and an adipogenic cocktail to induce differentiation over 48 h. The expression of PPARγ and C/EBPα was evaluated using qRT-PCR with specific primer pairs at the end of the differentiation process. The relative qRT-PCR values were corrected against β-actin expression levels and normalised with respect to the differentiation control (CTR). Data were obtained from at least 3 replicates from three independent experiments and are expressed as mean ± SD. The maximum mRNA expression levels at the end of the differentiation process for the untreated control cells (CTR) were set at 1, and relative mRNA expression levels in cells treated with CCT10μM at the same time point are depicted. (**C**) Oil Red O staining of 3T3-L1 adipocytes. The differentiation of 3T3-L1 preadipocytes to adipocytes was carried out for 6 days before being stained with OR. Representative images of adipocytes differentiated in absence (CRT) or presence of CCT (CCT10μM). Images were captured under an Olympus IX51 microscope (Olympus, Tokyo, Japan) using a 20X LCAch N, 0.4 NA objective and a Leica DC 500 CCD camera (Leica-microsystems, Wetzlar, Germany). Capturing conditions resulted in a pixel size of 2.9 micrometres. * and ^a^ indicate a significant difference compared with control differentiation or DMSO, respectively (*p* < 0.05).

## Data Availability

The data presented in this study are available on request from the corresponding author.

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
