# Peer review of "Effect of Crocetin on Basal Lipolysis in 3T3-L1 Adipocytes"

_antioxidants, 2023, doi:10.3390/antiox12061254_

Round 1
Reviewer 1 Report
The manuscript titled " Effect of Crocetin on basal lipolysis in 3T3-L1 Adipocytes" submitted to the journal "Antioxidants" by Cimas et al., explores the impact of Crocetin, a saffron-derived compound, on lipolysis in adipocytes. The study demonstrates that Crocetin decreases glycerol release, downregulates adipose tissue triglyceride lipase (ATGL) and perilipin-1, and exhibits antioxidant and anti-inflammatory effects. These findings suggest the potential of Crocetin as a promising biocompound for improving lipid mobilization in obesity. To enhance the clarity and impact of your manuscript, I recommend considering the following points:
– Clarify research objectives: Clearly state the specific research objectives or hypotheses that the study seeks to answer. This will help readers understand the purpose and direction of the research.
– Some parts of the introduction contain detailed information that could be presented more concisely. Consider condensing certain sections to improve readability and focus on the key aspects relevant to the study. Reduce the number of references, consider prioritizing the most important and influential references that directly support the key points and objectives of the study. Selecting a subset of highly relevant references can help streamline the introduction and make it more focused and concise.
– The authors should define gene names the first time they are mentioned in the manuscript to ensure clarity and understanding.
– In the "Materials and Methods" and "Results" sections, you can enhance the organization and readability by adding numbers to the subsections.
– In the Results section, you can consider omitting the mention of "error" after the mean value if the corresponding error information is already provided in the figures. This can help avoid redundancy and streamline the presentation of results
– In the Results section, it is recommended to present cell viability data before discussing glycerol release. This ordering allows for a logical flow of information, starting with an assessment of cell viability as a measure of cell health and integrity, followed by the analysis of glycerol release as an indicator of lipolysis.
– In the figures, it is recommended that the authors use clear and descriptive labels for the samples instead of abbreviations such as "CCT0uM." Instead, it would be more informative and reader-friendly to label the samples as "Control" or "Non-treated cells" to clearly indicate the respective groups being compared.
– The authors should include the control sample in Figure 3 to provide a comprehensive visual representation of the experimental groups.
– To obtain a comprehensive assessment of intracellular fat accumulation and the expression of PPARγ and C/EBPα, it is recommended to measure these parameters at the end of the differentiation process rather than only after a 48-hour treatment.
– I recommend focusing on improving the English language usage by paying attention to sentence structure, incorporating transitional phrases for better flow, maintaining consistent verb tenses, using precise vocabulary and terminology, and thoroughly proofreading and editing the manuscript to correct any grammatical errors or inconsistencies.
Reviewer 2 Report
The authors of the study titled “Effect of Crocetin on basal lipolysis in 3T3-L1 Adipocytes” investigated the effect of crocetin (CCT) on lipolysis, key lipolytic enzymes and on nitric oxide synthase (NOS) expression as well as lipid accumulation and antioxidant enzymes activities in differentiated 3T3-L1 adipocytes. My only comment will be:
a) Add a photo of adipocytes with and without CCT after Oil Red O staining showing lipid droplets in mature adipocytes
b) In discussion remove detailed information about mechanism of lipolysis which already are described in Introduction, the same with other doubled information
Reviewer 3 Report
The authors examined the anti-adipogenic and anti-oxidant activity of crocetin. An anti-lipolytic affect was suggested by the downregulation of ATGL and perilipin gene expression but unchanged HSL levels.
Anti-oxidant activity of crocetin was suggested by increased levels of catalase and SOD activity. However, why is it not possible that crocetin increases oxidative stress and increased levels of catalse and SOD are compensatory upregulated? It would be more informative if levels of oxidative stress markers would be quantified (malondialdehyde, levels of GSH) beside the activity of anti-oxidant enzymes.
Figure 3: panel A and C indicate catalase and GPx activity with nmol/min/ml, but in the Figure legend it is stateted that all measurements are shown as percentage relative to 0 µM CCT treatment: what is shown in panels A and C: percent activity relative to CCT0µM or activity in nmol/min/ml? (see also next comment:) I would expect that CCT0µM were cells treated with the solvent (DMSO) alone: but then it is not clear what is the difference between CCT0µM and DMSO?
The same applies to Figure 1 and Figure 6: CCT0µM and DMSO are shown separately: it is not clear what it the difference? The CCT0µM should have the same DMSO as the CCT10µM (otherwise "untreated control" would be a better description).
Line 385 and 387: it is stated that "ISO data panels are missing from Figure 4": why are they missing?
Minor points:
Figures 2, 4 and 5: 0uM and 10uM in the Figure panels should be replaced by 0µM and 10µM ("micro" instead of "u")
Round 2
Reviewer 1 Report
Thank you for revising your document based on the reviewer's comments. I appreciate the thoroughness with which you addressed the feedback. The revisions made have significantly improved the manuscript, resulting in a more robust and coherent piece of work.
Reviewer 3 Report
Most important critical points have been adressed/errors corrected;
Manuscript may be accepted after correcting the following error:
Figure 1, panel B: replace CCT10mM by CCT10 µM
